# Dinuclear vs. Mononuclear Copper(II) Coordination Species of Tylosin and Tilmicosin in Non-Aqueous Solutions

**DOI:** 10.3390/molecules27123899

**Published:** 2022-06-17

**Authors:** Ivayla Pantcheva, Radoslava Stamboliyska, Nikolay Petkov, Alia Tadjer, Svetlana Simova, Radostina Stoyanova, Rositza Kukeva, Petar Dorkov

**Affiliations:** 1Faculty of Chemistry and Pharmacy, Sofia University “St. Kliment Ohridski”, 1164 Sofia, Bulgaria; radoslava_d_dimitrova@abv.bg (R.S.); fhat@chem.uni-sofia.bg (A.T.); 2Institute of Organic Chemistry with Centre of Phytochemistry, Bulgarian Academy of Sciences, 1113 Sofia, Bulgaria; sds@orgchm.bas.bg; 3Institute of General and Inorganic Chemistry, Bulgarian Academy of Sciences, 1113 Sofia, Bulgaria; radstoy@svr.igic.bas.bg (R.S.); rositsakukeva@yahoo.com (R.K.); 4Research and Development Department, Biovet Ltd., 4550 Peshtera, Bulgaria; p_dorkov@abv.bg

**Keywords:** macrolide antibiotics, copper(II) complexes, spectral properties, DFT/PCM calculations

## Abstract

The veterinary 16-membered macrolide antibiotics tylosin (HTyl, **1a**) and tilmicosin (HTilm, **1b**) react with copper(II) ions in acetone at metal-to-ligand molar ratio of 1:2 to form blue (**2**) or green (**3**) metal(II) coordination species, containing nitrate or chloride anions, respectively. The complexation processes and the properties of **2**–**3** were studied by an assortment of physicochemical techniques (UV-Vis, EPR, NMR, FTIR, elemental analysis). The experimental data revealed that the main portion of copper(II) ions are bound as neutral EPR-silent dinuclear complexes of composition [Cu_2_(µ-NO_3_)_2_L_2_] (**2a**–**b**) and [Cu_2_(µ-Cl)_2_Cl_2_(HL)_2_] (**3a**–**b**), containing impurities of EPR-active mono-species [Cu(NO_3_)L] (**2a’**–**b’**) and [CuCl_2_(HL)] (**3a’**–**b’**). The possible structural variants of the dinuclear- and mono-complexes were modeled by the DFT method, and the computed spectroscopic parameters of the optimized constructs were compared to those measured experimentally. Using such a combined approach, the main coordination unit of the macrolides, involved in the complex formation, was defined to be their mycaminosyl substituent, which acts as a terminal ligand in a bidentate mode through the tertiary nitrogen atom and the oxygen from a deprotonated (**2**) or non-dissociated (**3**) hydroxyl group, respectively.

## 1. Introduction

Metal complexes of biologically active molecules continuously attract research interest due to the various effects that the metal ions can exert on the properties of the parent compounds. Metal homeostasis and the equilibria between the biometal ions and the native bioligands are the main processes responsible for the well-being of the organisms. These equilibria can be changed upon addition of therapeutics which may alter the physiological levels of metal ions and their subsequent activity due to the formation of new coordination compounds. In turn, the metal ions can also influence the in vivo activity of medications depending on their complexation ability and the origin of the binding sites, which may result further in the development of novel metal-based drugs. 

Among the widely studied cations are the copper(II) ions which, as essential, play a crucial role in metal homeostasis and are responsible for the normal biological activity. The coordination chemistry of this transition metal ion has received considerable attention due to its potential impact for bioinorganic chemistry. Copper(II) complexes were found to possess properties such as antimicrobial, antiviral and anti-inflammatory activity, some of which are more pronounced compared to those of the non-coordinated ligands [1,2,3,4,5].

The literature overview reveals that the coordination chemistry of copper(II) ions towards the macrolide class of antibiotics has not been profoundly explored. Discovered in the 1950s with the isolation of erythromycin, the number of natural and semi-synthetic macrolides constantly grows together with their application in human and veterinary medical practices [6,7,8,9,10,11,12,13,14,15,16,17,18,19,20]. Despite the different sizes of the lactone ring (14, 15 and 16 members) and the various functionalities substituted to it, all macrolides in clinical use share similar saccharide substituents attached to the 5-hydroxyl group-D-desosamine or β-D-mycaminose. Both saccharides contain a tertiary nitrogen atom and hydroxyl group(s) at close distance(s), capable of binding metal ions and forming 5-membered chelates. This N,O-bearing environment appears to be the main coordination site of the macrolides towards copper(II) ions and could play different roles in the formation of mono-, di- or polynuclear complex species depending on the reaction conditions applied [21].

The only previously reported interaction among the clinically used macrolides towards copper(II) ions is that of the 15-membered azithromycin (HAzi). The earliest investigation dates back to 1995 when the crystal structure of [CuAzi_2_] was reported, where two azithromycinates bind the metal(II) center through the tertiary nitrogen atoms and the oxygen atoms from the deprotonated hydroxyl groups, resulting in a very irregular structure of composition [CuN_2_O_2_] [22]. In addition, the authors discussed the formation of a significant cavity along the two-fold screw axis and its role in hosting a number of solvent molecules that prevent crystal decomposition. Sher et al. [23] evaluated the interaction between copper(II) ions and HAzi under equilibrium conditions to observe the formation of [CuAzi]^+^ and [CuAzi_4_]^2−^ complex species. Their findings revealed the predominant binding of the metal(II) center to the oxygen-donor atoms rather than to the N,O-coordination set. Twenty years later, a mononuclear copper(II) complex of composition [CuAzi_2_(H_2_O)_2_] × 2H_2_O was also proposed, bearing as main chromophore unit [CuN_2_O_2_] [24].

The lack of studies triggered our interest in the coordination of macrolides towards copper(II), especially in the ability of 16-membered macrolide antibiotics to bind Cu(II). The natural tylosin and its semi-synthetic derivative tilmicosin were selected as target ligands due to their definite advantages over the 14- and 15-membered erythromycin-based derivatives, e.g., better gastrointestinal tolerance, no drug incompatibility and activity against certain resistant bacterial strains [25]. Moreover, we recently reported the isolation and characterization of homoleptic mononuclear Cu(II)-bis-tylosinate and Cu(II)-bis-tilmicosinate, containing a [CuN_2_O_2_] coordination core [26]. In this case, the complexation reactions occur in aqueous solution in the presence of ligand excess, and the active site of macrolides is their mycaminosyl substituent. The formation of these complex species is in agreement with the known data on various mononuclear bis-aminoalcoholates obtained under similar reaction conditions. On the other hand, the aminoalcohol-bearing ligands are able to form di- or polynuclear coordination species in non-aqueous solutions [21], which in turn leads to the following general questions: How will the macrolides behave in the presence of copper(II) ions in such an environment? What is the coordination mode of the antibiotics in non-polar media? Can coordination species of different composition and structure co-exist?

In the present paper, we discuss our findings on the ability of tylosin (HTyl) and tilmicosin (HTilm) (Figure 1) to bind copper(II) ions in acetone solutions. The complexation processes and the properties of the novel metal(II)-based derivatives of both macrolides were evaluated by a combined approach, including experimental studies and theoretical calculations. 

## 2. Results and Discussion

The natural 16-membered macrolide antibiotic tylosin base (**1a**) and its semi-synthetic analog tilmicosin base (**1b**) interact with copper(II) nitrate or copper(II) chloride in acetone at metal-to-ligand molar ratio of 1:2. The resulted reaction mixtures were precipitated with diethyl ether or hexane to isolate the blue (**2**, Cu_x_L_y_(NO_3_)_z_)) and the green (**3**, Cu_x_L_y_Cl_z_) products in solid phase, respectively. As will be seen further, these products consist predominantly of neutral dinuclear copper(II) complexes (**2a**–**b**/**3a**–**b**), containing non-stoichiometric impurities of the corresponding mono-species **2a’**–**b’**/**3a’**–**b’**. The composition and structure of the di-and mono-complexes of copper(II) were derived using a number of spectral techniques and elemental analysis data along with density functional theory (DFT) modeling. 

### 2.1. Physicochemical Properties of the Blue (**2**) and Green (**3**) Copper(II) Complexes

The electronic spectra of the isolated blue (**2**) and green (**3**) copper(II) macrolidates in ethanol solution consist of an intense signal at 280–285 nm and a weak, broad band in the range from 650 to 850 nm (Figure 1 a,b). The high-energy absorbance is attributed to the π–π* transitions occurring in the macrolide rings of both ligands. Compared to the UV spectra of the native antibiotics, the signal position remains intact in the spectra of the copper(II) complexes, demonstrating that the tylactone ring does not interact with the metal(II) cations. The low-energy, almost symmetrical band in the spectra of the blue (650–720 nm) and green (730–850 nm) complexes most likely arises from d–d transitions in the copper(II) centers bound predominantly with oxygen donor atoms (**2**) or an O, Cl-containing set (**3**). The apparent significant absorption of the blue tylosinate in ethanol is due to its partial solubility in this solvent, which shifts up the baseline due to solution turbidity.

The replacement of ethanol with acetone (Figure 1 c,d) does not significantly affect the position and the shape of the bands in the blue complexes, although a hyperchromic shift is detected in the blue tylosinate. The use of acetone in the green complexes causes the following changes in their electronic spectra: (i) a significant batho- and hyper-chromic shift of the transitions in the tylosinate complex, accompanied by the appearance of a new, relatively intense signal at 475 nm and a shoulder at ca. 700 nm; (ii) an asymmetric multicomponent band for the tilmicosinate derivative, with a maximum at 755 nm and shoulders at 630 and 480 nm**,** respectively. Compared to ethanol, the following transitions, arising from the metal(II) ion interaction with different donor atoms, can be distinguished in the polar aprotic solvent: Cu–O (ca. 750–850 nm), Cu–N (635–700 nm) and Cu–Cl (charge transfer (CT), 475–480 nm). 

The electronic spectroscopy provides the first evidence that the blue and green solids comprise copper(II) complexes of tylosin (**1a**) and tilmicosin (**1b**). At this stage of research, due to the unknown composition of the coordination species **2** and **3** (mono- or di-), the extinction coefficients of the ligands, the blue/green products and the starting Cu(II) salts were calculated. Later, assuming that the dinuclear structures prevail, the corresponding molar extinction coefficients were also evaluated (Table 1). In all cases, the spectral behavior of the copper(II) ion placed in the corresponding crystal field differs from that observed for Cu(NO_3_)_2_ × 3H_2_O and CuCl_2_ × 2H_2_O dissolved in the same solvents.

Using the continuous variance method (a Job’s plot), the calculated metal-to-ligand stoichiometry in the isolated blue and green macrolidates was found to be 1:1. Representative Job’s plots of both copper(II) tilmicosinates are shown in Figure 2. The ratio between the metal(II) cations and the macrolide antibiotics was confirmed by the elemental analysis data (Table 2), which in addition indicate the presence of one mole of nitrate anions (blue complexes) or two moles of chlorides (green complexes) per one mole of copper(II) ions, respectively. The overall neutral character of the species formed is secured by the deprotonation of the antibiotics in the blue complexes and their participation in the non-dissociated form in the green coordination compounds [21]. The observed elemental composition can be attributed to the formation of two possible constructs which contain one (mononuclear species, **2a’**–**b’**/**3a’**–**b’**) or two (dinuclear complexes, **2a**–**b**/**3a**–**b**) metal(II) centers ligated by the corresponding number of antibiotic ligands and inorganic anions. Based solely on the absorption properties of the copper(II) complexes formed and the elemental analysis data, it cannot be inferred which species prevail, which is why we performed additional experimental studies to elucidate the composition and structure of the newly obtained blue and green macrolidates. 

The 16-membered veterinary antibiotics contain a mycaminose substituent, which was already found to be the main coordination site of the mononuclear copper(II)-bis-macrolidates, isolated under water–alkaline conditions [26]. The suitable position of the N- and OH-donor atoms in this sugar moiety (an aminoalcohol fragment) is a prerequisite for the formation of dinuclear Cu(II) complexes in non-aqueous solutions [21], which is why we assume that the blue and green coordination species contain a dinuclear chromophore and are of composition [Cu_2_(µ-NO_3_)_2_L_2_] (**2a**–**b**) and [Cu_2_(µ-Cl)_2_Cl_2_(HL)_2_] (**3a**–**b**), respectively. The suggested dinuclear complexes can be constructed by two metal centers bound with two bidentate aminoalcohol(ate) fragments, which in addition serve as a bridge between the copper(II) ions (Figure 2a,b). The other places in the inner coordination shell can be occupied by various anion/solvent combinations. The possibility for a geometric isomerization of such structures, where the bridging inorganic anions bind the two metal centers and the bidentate macrolides are terminally coordinated, cannot be totally excluded (Figure 2c,d). 

The bridging oxygen atom originating from the macrolide sugar fragment can technically be involved in O-Cu charge transfer transitions, observable in the 350–400 nm range [27,28,29,30,31,32,33,34,35]. The absence of such absorbance in the electronic spectra of the blue and green species raised several questions: (i) What is the coordination mode of the macrolides? (ii) Are the suggested dinuclear structures (**2a**–**b**/**3a**–**b**) preserved in solution? (iii) Can dissociation of the proposed dinuclear species to the corresponding mono-complexes (**2a’**–**b’**/**3a’**–**b’**) occur?

In order to dispel the above-mentioned doubts, we performed a series of EPR studies. The EPR experiments were performed on powder copper complexes and their frozen solutions in acetone and ethanol. Generally, the formation of bridged copper(II) structures leads to EPR-silent species due to the spin coupling caused by the close proximity of the two paramagnetic centers (ca. 3 Å).

For the copper systems studied by us, the EPR spectra of the powder samples **2** and **3** show an axially symmetric signal. The hyperfine structure is partly visible in the parallel component. The EPR parameters of the blue macrolidates are similar with g_II_ = 2.27, g_⊥_ = 2.06 and hyperfine structure constant A**_‖_** = 170 G. The values slightly deviate from those determined for the green samples: g**_‖_** = 2.26, g_⊥_ = 2.07 and A**_‖_** = 146 G (tylosin); g**_‖_** = 2.25, g_⊥_ = 2.05 and A**_‖_** = 164 G (tilmicosin). The experimental EPR parameters g_‖_ > g_⊥_ are typical for copper(II) ions placed in a tetragonally distorted octahedral crystal field with a single electron occupying the $d_{x^{2}-y^{2}}$ orbital (Table 3). The observed hyperfine structure arises from the interaction of the single electron (S = 1/2) with the nuclear spin of ^63,65^Cu (I = 3/2). The experimental, simulated and calculated EPR spectra of the blue and green species are presented in Figure 3 and Figure 4, respectively. The EPR parameters are summarized in Table 3.

The spin concentration in the studied samples was experimentally determined using the SpinCount program implemented in the Bruker EPR software. In Table 4, we present the theoretically expected number of the unpaired spins for the blue and green compounds with the experimentally observed spins both in powder state and in frozen acetone solutions, assuming that the samples comprise solely copper(II) mono-complexes. For the sake of comparison, the spin concentration of copper(II) nitrate and copper(II) chloride is also given. As one can see, there is a significant difference between the experimentally and theoretically calculated spin numbers of the target copper complexes irrespective of their aggregation state. This means that only a small part of the nominal copper content is detectable by EPR spectroscopy, which corresponds, most probably, to the presence of the EPR-active mononuclear species **2a’**–**b’** and **3a’**–**b’**. They co-precipitate in the solid phase, retaining the main binding features of the macrolide antibiotics and the corresponding inorganic anions, but this spontaneous process cannot be controlled, and the amount of mono-species was found to be non-reproducible in consecutive syntheses in contrast to their constant EPR parameters. 

The comparisons of the EPR parameters for powder mono-species and their acetone and ethanol solutions, as well as for the acetone and ethanol solutions of the parent copper(II) salts, reveal some coordination features of the copper(II) centers in the studied complexes (Table 3). The closeness of the EPR parameters observed for ethanol and acetone solutions of the mono-species **2a’** and **3a’** on the one hand, and copper(II) nitrate in the same solvents on the other hand, implies a weak copper(II)–tylosin bonding. On the contrary, the semi-synthetic antibiotic appears to form more stable copper(II) complexes, since identical EPR parameters were found for the tilmicosin-containing mono-species **2b’** and **3b’** both in solid state and in ethanol/acetone solutions. The values of the covalency factor α^2^ > 0.5 for all mono-complexes in solid state disclose the overall ionic character of the bonds formed. In addition, no strong exchange interaction between the metal(II) centers is observed for the mono-complexes **2a’**, **2b’** and **3b’**, as evidenced by the values of the parameter G > 4. Details on the calculation of α^2^ and G are presented in Section 3. 

The NMR studies of the isolated blue and green copper(II) complexes in acetone-d_6_ reveal that tylosin and tilmicosin are bound to the paramagnetic metal(II) center (Figure 5 and Figure 6). The signals of the skeletal macrolide protons remain relatively intact, although they broaden in the spectra of the copper(II)-containing species in comparison to the non-coordinated ligands. The protons of the N,N-dimethyl groups (ca. 3 ppm) considerably change their chemical shifts to a lower field, indicating the place of the metal(II) ion coordination. These observations are in agreement with the already witnessed experimental data that the mycaminose sugar plays a key role in coordinating copper(II) ions [26]. The more detailed interpretation of the proton NMR spectra of the complexes could shed light on some intrinsic specificity of the ligand arrangement in the dinuclear complexes. Our principal task in the present research is to evaluate the primary coordination sites of tylosin and tilmicosin, and for that reason, we did not go further into the particular NMR properties of these species. The unambiguous assignment of all the signals also requires ^13^C-NMR studies, but due to the presence of paramagnetic ions, the carbon signals are quite broad, providing no additional structural information on the macrolide binding. Nevertheless, the diffusion-ordered spectroscopy (DOSY) studies confirm that the main portion of the antibiotics remains in the form of dinuclear species exhibiting, e.g., the values of 1.09 × 10^−9^ and 0.56 × 10^−9^ m^2^·s^−1^ for the diffusion coefficients of tilmicosin and its green complex, respectively, in acetone solutions. 

In summary, for the first time, we report the isolation and the spectral characterization of four novel dinuclear Cu(II) complexes of tylosin and tilmicosin. The experimental study demonstrates that the coordination of the essential copper is a complicated process, which depends on the reaction conditions, the nature of the copper(II) salt and the molar metal-to-ligand ratio. Without having suitable single crystals for X-ray crystallographic studies, it is speculative to discuss the precise coordination mode of tylosin and tilmicosin. However, as will be seen further, by combining the experimental data with quantum chemical modeling, we are able to put forward the most reliable structures of the complexes formed.

### 2.2. Quantum Chemical Calculations

In our previous study, the mycaminosyl moiety was proved to be the sole fragment in the macrolide molecules that takes part in the formation of the first coordination shell in the mononuclear complexes [CuTyl_2_] and [CuTilm_2_] [26]. In the present study, the spectroscopic data are also in agreement with such a conjecture, and on this basis, using the density functional theory (DFT), we designed various dinuclear constructs bearing two metal centers and two mycaminose saccharides. In order to explain the presence of the EPR signals belonging to the isolated copper ions, the structures of the suggested mono-species were evaluated as well. 

Taking into account the possible complexation patterns of the dinuclear complexes containing nitrate anions (**2a**–**b**), two general structures were modeled, where the nitrate ligands play a bridging role (**4**) or are placed at terminal positions (**5**), respectively (Figure 7). The environment of the individual copper(II) ions in compound **4** appears to be a square planar geometry achieved by two µ-nitrates and two bidentate deprotonated macrolides. The trigonal bipyramid in complex **5** arises from the interaction of each metal(II) center with a bridging bidentate antibiotic anion and a η_2_-nitrate. 

The calculations indicate that the most probable structure of these species consists of two bridging nitrate anions and two deprotonated mycaminosyl substituents serving as terminal ligands (**4**). The energy difference between the constructs containing terminal (**5**) and bridging (**4**) inorganic anions optimized with B3LYP is +489.1 kcal/mol in vacuo and +493.7 kcal/mol in implicit acetone, respectively. This significant contrast could be explained by the possible steric hindrance between the ligating N,N-dimethylamino groups and the terminal nitrates, leading to the conclusion that the only populated structure responsible for the spectral behavior of the blue complexes comprises nitrate ions linking the two metal cations (**4**). Thus optimized, the structure of the blue macrolide complexes consists of two copper(II) ions, each placed in a coordination environment, realized by the terminal bidentate N,O-mycaminosyl fragment and the monodentate bridging inorganic anions.

The data in Table 1 imply that the longest wavelength absorption is most likely due to a d–d transition, indicating the character of the frontier orbitals. Assuming that two copper(II) centers, each with an odd number of electrons, are involved in the blue complex formation, a question of the multiplicity of the resulting species arises. There are three possible spin configurations (Figure 8). Which one is dominant depends on the interplay between the energy difference ΔE_d–d_, i.e., the ligand field strength, and the exchange electron–electron interaction energy. The fact that the longest wavelength absorption is allowed suggests that it is a low-energy S_0_–S_1_ transition. Therefore, the theoretical modeling of the nitrate-bridged structure was further extended to shed light on the complex spin state—closed-shell (a) or open-shell (b) singlet. 

After optimization in vacuo, the closed-shell singlet construct possesses higher energy (+28.4 kcal/mol), so the open-shell singlet is the spin state of the complex in vacuo. Yet, the difference between the two spin states in acetone is less than 0.01 kcal/mol, indicating that these two singlet states are degenerate in solution and ΔE_d–d_ is relatively small. This is supported by the very good agreement of the calculated (Table 5) and measured (Table 1) absorption. 

The images of the frontier orbitals visualize the fact that the two quasi-degenerate SOMOs are practically of d-type (Figure 9). The orientation of the p-lone pairs of the nitrate oxygens is twisted with respect to the d-AOs, which results in the smallest d–d splitting (compared to the σ- or π-type overlap [36]) and explains the open-shell spin configuration.

The structural data as well as the spin density and the charge distribution of the nitrate-bridged complex **4** are presented in Figure 10. The results reveal a distorted rhombic environment around each Cu(II) ion. The shared nitrate oxygens and the amine nitrogens have identical charges and in effect identical distances from the metal centers, while individual oxygens are placed at a shorter distance and bear a higher charge. The shared oxygens shield the repulsion between the copper ions allowing them to be as close as 3.22 Å apart, which is an intermediate value between the lattice parameter of copper (3.61 Å) and the shortest Cu–Cu distance (2.56 Å) at room temperature [37]. The spin density distribution discloses that the spin of each Cu(II) ion is fairly delocalized, predominantly and unequally on the individual centers in the coordination shell. The spin density on the bridging oxygens is an order of magnitude smaller than the individual ones and could be ignored.

As discussed above, the experimentally observed EPR signals for the blue complexes **2a**–**b** were attributed to the presence of the corresponding mono-species **2a’**–**b’**; therefore, their structure was also modeled. The similarity of the EPR parameters allows inferring that they have identical structures for both ligands. In this case, the deprotonated mycaminose fragment and the nitrate anions act as bidentate ligands, but the two N,O- and O,O-donor sets lie in the same plane with the copper(II) cation (**6**, Figure 11) in contrast to the optimized nitrate-bridged framework. Nevertheless, the structural parameters and charges, even though calculated with a different computational protocol, are fairly close to those of the dinuclear construct. The theoretically calculated EPR parameters (Table 6) corroborate well the experimental data observed in solid state. 

In contrast to the blue complexes **2a**–**b** and according to the experimental data, the hydroxyl group from the mycaminose fragment remains non-dissociated in the structure of the dinuclear green complexes **3a**–**b** [21], and the inner coordination shell encompasses four chloride anions. The construction of the possible structural variances of the green complexes is challenging due to the multiple positions which can be occupied by the inorganic anions (mono- or bidentate; bridging/terminal). However, the most energetically favorable one (**7**) contains two bidentate N,OH-mycaminoses and two chlorides as terminal ligands, while the other two halogenide ions play a bridging role (Figure 12). Similar to the nitrate-bridged complex, the copper(II) centers are placed in a symmetrical coordination environment.

For this structure the two above-mentioned possible singlet states—open-shell and closed-shell—were also modeled in vacuo with B3LYP; the energy difference is 32.1 kcal/mol in favor of the open-shell singlet. When the acetone effects were taken into account, this difference reaches even the higher value of 35.7 kcal/mol, which unequivocally proves that the most populated state in both phases consists of two unpaired electrons. The calculated absorption energies for the open-shell singlet are in good correspondence with the experimentally observed values (longest calculated wavelength transition energy in vacuo is 1.29 eV (963 nm), f = 0.0004). The mutual orientation of the metal d-orbitals and the lone pairs of the ligands is as in **4** (Figure 13). Only one of the bridging chloride ions has a minor contribution to SOMO_1 and SOMO_2.

The structural parameters, spin density and charge distribution in the dinuclear green construct **7** are presented on Figure 14. Each copper center is placed in a distorted square-pyramidal coordination, the distances to the ligands being larger than those in the blue construct **4**. Yet remarkably, the Cu–Cu distance is exactly the same although copper charges are substantially lower. The charges of shared and individual chlorides are closer compared to the respective oxygens in the blue mycaminosate **4**. However, all of them are less than –1, which demonstrates their role as ligands, rather than as counterions. The spin density is more uniformly distributed but differs in pattern—largest on the individual chlorides, slightly lower on the shared ones and lowest at the oxygens. Apparently, the non-dissociated OH group has a different coordination behavior.

The corresponding mono-species derived from the dinuclear green complexes deserve more attention because of their different EPR parameters (Table 3). It might be assumed that, depending on the ligand (tylosin/tilmicosin), two types of mono-complexes bearing different donor atom-sets form. Due to its small size and high charge density, copper tends to form complexes with ligands bearing atoms of high electronegativity and propensity to form anions. In tilmicosin, the bulky substituted piperidine fragment is the closest additional ligating option, containing a nitrogen of lower electronegativity and steric inaccessibility, while in tylosin the oxygen from the aldehyde group could be part of the coordination shell. Therefore, two mono-nuclear structures were modeled: Without (**8a**) and with (**8b**) participation of the aldehyde group (simplified as an acetaldehyde). It is well known that the increased number of oxygen atoms occupying the first coordination shell leads to an increase in g_‖_ and a decrease in A_‖_, which is why the observed difference in the EPR parameters of **3a’**–**b’** could be attributed to the coordination of an additional O-donor at the axial position in species **3a’**. The optimized structures of two possible constructs are presented in Figure 15. The calculated data for the EPR parameters are summarized in Table 7. The good agreement with the measured data validates the suggested mononuclear structures.

In conclusion, the most adequate (energetically favorable) structures of the dinuclear copper(II)-based macrolides are derived by the theoretical modus operandi. The proposed constructs comprise two closely spaced paramagnetic(II) centers, where the open-shell singlet state is responsible for their EPR silence. The dinuclear coordination compounds contain a non-stoichiometric impurity of EPR-active mono-species, whose structures are also modeled. 

## 3. Experimental Section

### 3.1. Materials

The macrolide ligands—tylosin (containing > 90% tylosin A) and tilmicosin—were kindly provided by Biovet Ltd., (Peshtera, Bulgaria). The copper(II) salts and the solvents of analytical grade were purchased from local suppliers. Deionized water (18 MΩ·cm) was used in all experiments.

### 3.2. Methods

The microanalyses were carried out by the microanalysis service of the Institute of Organic Chemistry with Centre of Phytochemistry, Bulgarian Academy of Sciences (Sofia, Bulgaria). A Shimadzu UV-1800, 200–1000 nm (Kyoto, Japan) and a Nicolet 6700 FT-IR, 4000–400 cm^−1^, (Thermo Scientific, Madison, WI, USA) were used to record the UV-Vis and IR spectra, respectively. 

The registration of the X-band EPR spectra (77–295 K) was performed on a Bruker BioSpin EMX^plus^10/12 EPR spectrometer (Karlsruhe, Germany) working at 9.4 GHz. The values of the isotropic g-factor (g_iso_) and the parameters α^2^ (indicating covalent/ionic bond character) and G (accounting for exchange interaction) were calculated as follows [38]:$$g_{\mathrm{iso}}=\frac{g_{||\;}+2g_{⊥}}{3}$$
$$\alpha^{2}=\frac{A_{\left|\right|}}{P}+\left(g_{\left|\right|}-2.0023\right)+\frac{3}{7}\left(g_{⊥}-2.0023\right)+0.04$$
$$G=\frac{g_{\left|\right|}-2}{g_{⊥}-2},$$
where P = 386 G is the calculated anisotropic hyperfine structure constant of copper(II) ions in gas phase.

The ^1^H NMR spectra (600.13 MHz) of the isolated copper(II) complexes in acetone-d_6_ and the DOSY experiments at 293 ± 0.1 K were recorded on an AVANCE AV600 II^+^NMR spectrometer (Bruker, Karlsruhe, Germany).

The data for the elemental composition were adopted from the elemental analysis. Both terminal and bridging isomers were optimized in vacuo and in implicit acetone using the polarizable continuum model (PCM) [39,40] and utilizing two functionals with different percentages of exact exchange—B3LYP with the 6-311G(d) basis set and BHandHLYP with the cc-pVDZ one. Absorption energies of the dinuclear complexes were calculated with TD-BHandHLYP/cc-pVDZ. Optimized geometries and vertical transition energies were calculated with the Gaussian software package (Wallingford, CT, USA) [39].

Absorption energies of the dinuclear complexes were calculated with TD-B3LYP/6-311G(d)/PCM in implicit acetone. For the structures optimized in vacuo, the EPR parameters for the mononuclear particles were calculated using the ORCA software package with BHLYP/6-311G(d)/Wachters+f for copper [41,42]. For the calculations of the hyperfine structure constants, the Fermi contact hyperfine coupling, dipolar hyperfine coupling and orbital contribution to the hyperfine coupling were taken into account.

### 3.3. Synthesis

#### 3.3.1. Synthesis of Nitrate-Containing Cu(II) complexes, **2a**–**b**

First, 0.250 mmol HL (HL = HTyl, HTilm) was dissolved in acetone. Next, while stirring, an acetone solution of Cu(NO_3_)_2_ × 3H_2_O (0.125 mmol, 30.20 mg) was added. For isolation purposes, the reaction mixtures were precipitated with either hexane or diethyl ether. The blue precipitates obtained were then filtered off and dried over P_2_O_5_. Yield: 89% (**2a**), 93% (**2b**). Complex **2a** is insoluble in ether and hexane; it possesses a limited solubility in ethanol and acetone. Complex **2b** is insoluble in ether and hexane but is soluble in ethanol and acetone. 

#### 3.3.2. Synthesis of Chloride-Containing Cu(II) complexes, **3a**–**b**

First, 0.250 mmol HL (HL = HTyl, HTilm) was dissolved in acetone and while stirring, an acetone solution of CuCl_2_ × 2H_2_O (0.125 mmol, 21.25 mg) was added. The complexes were isolated as solids by precipitation of the reaction mixtures in either hexane or diethyl ether. The green precipitates obtained were then filtered off and dried over P_2_O_5_. Yield: 72% (**3a**), 84% (**3b**). Complex **3a** is insoluble in ether and hexane; it possesses limited solubility in ethanol and acetone. Complex **3b** is insoluble in ether and hexane but is soluble in ethanol and acetone.

## 4. Conclusions

The coordination behavior of the essential copper(II) towards the veterinary 16-membered macrolides tylosin and tilmicosin, investigated earlier in an aqueous solution with basic character [26], is supplemented with new data in a different synthetic setup using a solvent of decreased polarity (acetone). The resulting products were explored by a combined approach, employing an assortment of spectral techniques and molecular modeling at the quantum-chemistry level, to unveil the driving forces behind the structural and electronic features of the obtained complexes. The results are important first of all because they (i) validate the fact that in both mono- and dinuclear copper(II) complexes the coordination shell encompasses primarily the same fragment, the mycaminosyl moiety in a N,O-bidentate mode, irrespective of the remaining reaction conditions, and (ii) demonstrate that the pattern of complexation depends on the dielectric permittivity of the environment—mononuclear complexes in aqueous alkaline medium and dominance of dinuclear species in media of lower polarity. The bridging role in the dinuclear complexes is performed by the copper salt anions. The unpaired spins on the copper(II) ions are in an open-shell singlet configuration, interacting indirectly via the nonbonding orbitals of the ligating atoms, which is verified both by the computations and the EPR measurements. The different polarity of the experimental environment could be correlated with the conditions in different parts of the organisms, allowing a reliable prognosis of the behavior and activity of the complexes in bio-systems. Our preliminary in vitro assay reveals that the coordination of copper(II) ions enhances the effect of the native antibiotics, which is why a detailed study on the impact of Cu(II) ions on the bioactivity of 16-membered macrolides is underway.

## Data Availability

Data are available from the authors upon request.

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
