# Peer review of "Dinuclear vs. Mononuclear Copper(II) Coordination Species of Tylosin and Tilmicosin in Non-Aqueous Solutions"

_molecules, 2022, doi:10.3390/molecules27123899_

Round 1

Reviewer 1 Report

The manuscript by Ivayla Pantcheva and co-workers reports the copper(II) complexes of two antibiotics in acetone. The synthesized complexes were analyzed with UV-vis, EPR and NMR spectroscopic methods, moreover, DFT calculations were carried out. The presented work looks interesting; however, it lacks impact and scope. That I really miss the point, for which the reported copper(II) complexes should be given attention. What are the medical purpose to apply such copper(II) - antibiotics complexes? This part should be seriously strengthened. Overall, the standard of the work still needs to be improved to publish in Molecules. Hence, I am unable to recommend the work to publish in this journal and it can be accepted for publication after a carefully reevaluation.

Further comments:

(i) “The complexation processes occur in acetone at metal-to-ligand molar ratio of 1:2” This statement is confusing. The synthesis was carried out by using a 1:2 metal to ligand ratio and the authors reported good yields. On the contrary, the Job’s method (data should be reported) indicated 1:1 metal to ligand ratio and this is consistent with the results of elemental analysis.

(ii) The extinction coefficient values should be reported in M-1cm-1 unit. This allows better comparison with literature data.

(iii) In some cases, significant absorption (Abs ca. 1) was registered at 1000 nm. Is this due to the precipitation?

(iv) Did the authors check the stability of the copper(II) complexes?

(v) The EPR spectra should be fitted on the basis of the estimated g and A tensors. I do not understand the Cu:L ratios in the Table 3. If the authors used 65-fold excess of the ligands, the formation of dinuclear species should be suppressed.

(vi) In the DFT calculations, the authors used the saccharide part of the ligands to calculate the structure of the copper(II) complexes and their electronic features. Therefore, these complexes cannot be 2a,2b, etc. since they are only model systems. The corresponding Cartesian coordinates with the energy values and the number of imaginary frequencies should also be reported.

(vii) The authors reported good agreement between the calculated and measured absorption energies. After the comparison of the values reported in Table 1 and 4, this statement is not obvious. The authors should provide the calculated UV-vis spectra, not only the absorption energies.

Reviewer 2 Report

The submission by Pantcheva and co-authors concerns the characterization of Cu(II) coordination complexes with two macrolide antibiotics generated in acetone and characterized in non-aqueous solvents; this is a follow up to a previously report on quite similar complexes that were generated in basic aqueous solutions. One major difference is that in this case the dominant species appear to be dimeric (as per loss of EPR signal), rather than the monomeric complexes described in the previous report.   The syntheses and characterizations are fairly straightforward, very similar to previous report, but the limited methods used leaves structural ambiguity that xtallography or XAS might have resolved.  As in the previous report, the authors rely on DFT modeling to differentiate between possible structures.  Missing from both reports is any mention of binding affinities of Cu(II) for the macrolides, an important aspect of metallo-drug interactions in vivo, and the major justification in the introduction.

The work is competently done, but the presentation has several issues to be addressed.  

There are a number of awkward wordings which need to be copy-edited, e.g.

15 “veterinary 16-membered” rather than “veterinary antibiotic”;

41,42 – “The complexation and homeostasis of copper(II) ions have been widely studied due to their crucial role in normal biological activity”

47 -49 – “A literature review reveals a lack of reports on copper(II) interactions with macrolide antibiotics”

97 The Results and Discussion start without describing how the complexed samples 2a/b and 3a/b were generated and jump into how the physico-chemical characterizations. This confuses the reader and is easily addressed in rewrite.

Figures 1 is never explicitly mentioned in the text, it should be referenced in the discussion of results and interpretations.  In Figure 1, the complex concentrations are given as g/L instead of molarity, which is quite awkward for a chemical report.  This again added confusion as there was no clear description of the synthesis and isolation of the various compounds.

140 The text mentions “a Job plot”, should be Job’s plot, which gives ratios of metal/ligand as a first assessment of complex stoichiometry. This plot should be shown in the text, to verify the 1:1 stoichiometry.

171- “spin coupling” instead of “possible electron exchange”

179-180 tetragonal, not tetrahedral… better wording would be a “tetragonally distorted octahedral”

The authors assumption of spin-coupled Cu2 dimers should be proven. Analyte concentrations in EPR can be quantified by integration; the authors should show the expected intensity using a Cu(II) standard and estimate the concentration of the monomeric impurities.

207-212   The authors refer to alpha squared and G values, implying they describe the covalency in the species.  Both need to be explicitly defined and the rational used need to be described and referenced (more info on both are given in Methods, but the first use should be explained and referenced). Likewise, they mention temperature dependence of signal in powdered EPR but do not show the data or analysis, but then list undefined Weiss constants.  If these results are important, the method should be referenced and defined, the data should be shown and then the conclusions given.

Figures 3 and 4 show NMR comparisons of free and copper-complexed macrolides, concentrations should be given in the captions.  Again, authors describe a titration of ligands w/ “metal(II) salts” (Cu(II)?), in which the dimethylamine signals shift to lower field. This implies equilibration of bound and unbound macrolides. Again, data discussed should be shown (or not discussed).

255 Figure 5 and on,  the color scheme of the DFT figures is not given, ie, purple Cu, blue N, etc. Would be much easier if atoms were labeled in each figure. The text does not describe the coordination environment of individual copper ions, which appears to be 5 coordinate trigonal bipyramidal in the terminal and square planar in the bridging case. Normal inorganic designations would be that the former has an eta-two nitrate, while the latter has a mu-bridging nitrate. The dimetallic structure for the green dichloride modeled in Figure 10, the coordination would be square pyramidal, with two mu-bridging chlorides.

270-279  The discussion is confusion. The Cu absorbances from d- d transitions are inherently forbidden, that is why they are of such low intensity.  The breaking of pure octahedral symmetry increases the intensity, which predicts the “green” species proposed to be 5-coordinate square pyramidal should be significantly high intensity, data seems to show minor increase but not enough to rule out 6-coordinate distorted octahedral.  

Regarding spin configurations, Figure 6, did the authors use an EPR which can characterize an open shell S = 1 species, most X-band EPRs can’t.  All three configurations are accessible in many systems, dependent on the temperature; the spin coupling constant J is obtained by a temperature dependent change from one to the other.   

292-301  and. The discussion of the d-d character of absorbances is typically illustrated with a d-orbital splitting diagram, rather than a colored picture of the SUMO/LUMO orbital.  The two SOMOs shown in Figure 7 appear to be mixes of x2-y2 and z2 d-orbitals on the two copper ions, which seems unusual as simplistically for a square planar d9 complexes would expect SOMO to be x2-y2 only. Perhaps the authors might comment on this.  Similar arguments can be made for the green dinuclear structure, except the d-orbital splitting between x2-y2 and z2 orbitals is much smaller in square pyramidal complexes.

339 the coordination of protonated alcohol is quite rare, a reference should be used to discuss other reported instances

383 use of colon not necessary

400  use “modeled” instead of “designed” 

Recommend minor revision

Reviewer 3 Report

The synthesis, analysis, and quantum-chemical calculatiuons of two new dinuclear copper(II) complexes with the veterinary 16-membered macrolide antibiotics tylosin and tilmicosin is reported in the present article. This work is principally interesting, because the obtained compounds potentially can possess antimicrobial, antiviral, and anti-inflammatory properties. However, the work contains serious flows in data analysis; most of the conclusions about their structure are speculative. In my opinion, authors should carefully revise the manuscript and significantly improve the quality of data analysis.

I have the following concerns and suggestions:

1)            Table 1. The extinction coefficients are usually given in L mol-1cm-1 (M-1 cm-1)  units. I suggest you to convert extinction coefficients from L.g-1 .cm-1 to L mol-1cm-1 (M-1 cm-1) units.

2) Lines 182-184: Authors stated that “The low intensity of the EPR signals concomitant with the calculated number of paramagnetic centers in 2a-b and 3a-b indicate that the registered spectra belong to monomeric impurities (2a’-b’ and 3a’-b’, respectively, Figure 2, Table 3).” Please, describe how did you make such conclusion. Currently, the abovementioned statement looks postulated but not concluded from Fig. 2 and Table 3.

3) Lines 214-223 and Figure 4: The NMR spectra are almost not discussed. One can observe in Fig. 4 that spectra of ligands (Tylosin and Tilmicosin bases) and the copper complexes look similar, but have a lot of differences when analyzed more carefully. Please, extend the discussion of NMR spectra, because they probably contains important structural information.

4) It is difficult task to distinguish between open-shell and closed-shell singlet state by DFT methods. How did you find that calculated structure of copper(II) dinuclear complexes 1a, 3a, 2b, 3b is open-shell but not closed-shell singlet? What is the energy difference between open-shell and closed-shell singlet molecules?

5) Lines 293-301: The authors assigned the transition from SUMO to LUMO orbital as d-d-transition based on the analysis of molecular orbitals. However, Cu2+ ion has 3d9 configuration and there is no possibility to have unoccupied d-orbital. d-d (or so-called Ligand Field) transitions of copper(II) compounds involves transition of electrons from lower-lying fully occupied d-orbitals to  single occupied (SUMO) d-orbital.

Round 2

Reviewer 1 Report

The authors have corrected the major and minor indications suggested earlier. I consider this manuscript adequate to be published in the present form and I recommend it for publication.

Reviewer 2 Report

47-49  suggest     A literature review reveals that the coordination chemistry… (remove “although intensely studied,)

60- suggest    the only previously reported (replace studied)

74 suggest   The lack of studies triggered our interest in the coordination of macrolides towards copper(II), especially in the ability of 16-membered macrolide antibiotics to bind Cu(II).  

86  suggest to replace “aminoalcohols” with more commonly used “alkanolamines”

114  (Figure, top)  124 (Figure, bottom) same for other figures and schemes

119 suggest   most likely

121-3  turbidity results in scattering of the incident light, which is wavelength independent.  A simple solution is to find a null absorbance wavelength, one at which absorbance is expected to be close to zero (I suggest 1000 nm), and then subtract the absorbance at that point from the whole spectrum… basically a background correction.

124 suggest: Solvation by ethanol rather than acetone does not significantly affect…

181 suggest   Both 16-membered veterinary antibiotics studied contain….

Table 4, gives spin concentrations theory vs data; this also suggests the % of Cu monomer out of total. Please comment on this, as it suggests equilibrium.

278  “remains in the form of dinuclear species observing e.g. the values”  I suggest giving the data first, then interpreting.  “the diffusion constants determined by DOSY, xx yy, confirm that the …”

287 remove “as will be seen further,”

Reviewer 3 Report

The authors carefully revised the manuscript by addressing my comments. The authors significantly improved the quality of the data presentation and discussion of the data. Currently, the manuscript can be accepted in the present form.